**Data Availability Statement:** All data relevant to the study are included in the article or uploaded as supplementary information. The interview guide for the participants has been included as a

# Acceptability of a nurse-led non-pharmacological complex intervention for knee pain: Nurse and patient views and experiences

Polykarpos Angelos Nomikos[1,2,3]*, Michelle C. Hall[2,3,4], Amy Fuller[1,2], Reuben Ogollah[5], Ana M. Valdes[1,2,3], Michael Doherty[1,2,3], David Andrew Walsh[1,2,3], Roshan das Nair[3,6,7], Abhishek Abhishek[1,2,3]

1 Academic Rheumatology, Division of Rheumatology, Orthopaedics and Rheumatology, School of Medicine, University of Nottingham, Nottingham, United Kingdom, 2 NIHR Nottingham Biomedical Research Centre, University of Nottingham, Nottingham, United Kingdom, 3 Pain Centre Versus Arthritis, University of Nottingham, Nottingham, United Kingdom, 4 School of Health Sciences, University of Nottingham, Nottingham, United Kingdom, 5 Nottingham Clinical Trials Unit, University of Nottingham, Nottingham, United Kingdom, 6 Institute of Mental Health, Nottinghamshire Healthcare Foundation NHS Trust, Nottingham, United Kingdom, 7 Division of Psychiatry & Applied Psychology, School of Medicine, University of Nottingham, Nottingham, United Kingdom

* Polykarpos.nomikos@nottingham.ac.uk

## Abstract

### Objectives

The overall purpose of this research programme is to develop and test the feasibility of a complex intervention for knee pain delivered by a nurse, and comprising both non-pharmacological and pharmacological interventions. In this first phase, we examined the acceptability of the non-pharmacological component of the intervention; issues faced in delivery, and resolved possible challenges to delivery.

### Methods

Eighteen adults with chronic knee pain were recruited from the community. The intervention comprised holistic assessment, education, exercise, weight-loss advice (where appropriate) and advice on adjunctive treatments such as hot/cold treatments, footwear modification and walking aids. After nurse training, the intervention was delivered in four sessions spread over five weeks. Participants had one to one semi-structured interview at the end of the intervention. The nurse was interviewed after the last visit of the last participant. These were audio recorded and transcribed verbatim. Themes were identified by one author through framework analysis of the transcripts, and cross-checked by another.

### Results

Most participants found the advice from the nurse easy to follow and were satisfied with the package, though some felt that too much information was provided too soon. The intervention changed their perception of managing knee pain, learning that it can be improved with

1 / 14

supplementary file. The interview guide for the nurse is included as supplementary file in our published fidelity study. Additional qualitative data and audio recordings are archived in the University of Nottingham servers using password protection. Please email the Musculoskeletal Project Manager for the NIHR Nottingham Biomedical Research Centre at Bonnie.Millar@nottingham.ac.uk for requests to access this additional data.

**Funding:** This work was supported and co-funded by the NIHR Nottingham Biomedical Research Centre and the Pain Centre Versus Arthritis (Internal funding 2017- 2022).The views expressed are those of author(s) and not necessarily those of the NHS, the NIHR or the Department of Health and Social Care. The study is sponsored by the University of Nottingham, UK.

**Competing interests:** The authors have declared that no competing interests exist.

self-management. However, participants thought that the most challenging part of the intervention was fitting the exercise regime into their daily routine. The nurse found discussion of goal setting to be challenging.

## Conclusion

The nurse-led package of care is acceptable within a research setting. The results are promising and will be applied in a feasibility randomised-controlled trial.

## Introduction

Knee osteoarthritis (OA) affects 16% of the general population worldwide [1] and its' management remains challenging in most healthcare systems due to the sheer disease burden and limits on resources. Patient education, exercise, and weight loss where appropriate are recommended as core treatments for OA [2]. However, these interventions are often underutilised because of physicians' knowledge gaps, other demands on their time, and undue emphasis on drugs [3–8].

In the current model of care, patients with symptomatic knee OA may consult a General Practitioner (GP) or hospital specialist and be referred to a physiotherapist and/or dietitian for exercise and weight-loss advice as appropriate. Whether nurses can be trained to effectively deliver a complex non-pharmacological intervention for knee OA that includes components that are traditionally delivered by other allied healthcare professionals is yet to be determined. This is likely to be possible as previous studies have demonstrated efficacy of nurse-led care over usual GP-led care for chronic conditions when following a protocol, e.g., type II diabetes, heart failure, hypertension and gout [9–15]. Apart from this, it is not known whether nurse-led holistic care of knee OA would be acceptable to patients (as they would normally expect to be treated by different healthcare professionals for different aspects of their care), and, whether the nurse delivering such diverse interventions would find it acceptable to do so [16]. Patients are more likely to follow treatment recommendations if the intervention is acceptable [17, 18]. Similarly, practitioners only implement an intervention as intended if they find it acceptable to deliver [19].

Thus, the overall purpose of this study was to test the acceptability of a nurse-led non-pharmacological package of care for knee pain due to OA. The specific objectives were: to explore the experiences of participants who received the nurse-led non-pharmacological package of care for knee pain; explore the experiences of the nurse in delivering the intervention; and resolve possible challenges to future delivery. The present study forms part of a wider programme of work [20] that aims to evaluate the feasibility of a large randomised-controlled trial (RCT) for nurse-led package of care for knee pain due to OA.

## Methods

### Study design

Single arm mixed-methods feasibility study including quantitative fidelity assessment which has been previously reported [21] and qualitative acceptability assessment which is reported here.

### Ethical approval

The study received ethical approval by the East Midlands-Derby Research Ethics Committee (REC) (18/EM/0288).

## Consent

All participants gave written informed consent prior to their participation in the study.

## Patient and Public Involvement (PPI)

Three PPI members with hip or knee OA provided input into the content of the non-pharmacological treatment package, and volunteered for participating in nurse training sessions.

## Recruitment and participants

Participants of the Investigating Musculoskeletal Health and Wellbeing cohort [22] self-reporting knee pain and willing to receive information about future research studies were mailed a brief questionnaire about knee pain and an invitation to participate in the package development phase. Respondents to this questionnaire, aged 40 years and over, self-reporting knee pain for longer than 3 months and pain in or around the knee on most days of the previous month. Participants with at least moderate pain in two of the five domains of the Western Ontario and McMaster Universities Osteoarthritis Index (WOMAC) pain scale were invited to participate in the study [23]. Study participants were also limited to individuals who can read and write in English.

## Setting

Single centre secondary care study with community-based recruitment. All interviews were conducted in a private room in Academic Rheumatology, City Hospital, Nottingham. Where participants were unable to attend the site, telephone interviews were conducted.

## Intervention

The intervention consisted of a holistic assessment; education and advice; prescription of aerobic and strengthening exercise and weight-loss advice where required as recommended by NICE guidelines and is described in more detail in the protocol [20]. Specific advice around the use of adjunctive treatments such as the application of heat or cold, footwear modification and the use of walking aids were included although not allocated to individual sessions. Rather advice was tailored to the participants and the nurse advised on the wearing of footwear with a cushioned sole and the use of walking stick where participants walked with an antalgic gait. This is a more concise description of the intervention and further details are available in the study protocol [20]. Behavioural strategies were employed to motivate participants and improve adherence [24] these included SMART goal setting, action planning, assessment of participant confidence to achieve goals, discussion of barriers and facilitators and the use exercise diaries to improve adherence. The nurse delivered the non-pharmacological package of care over a five-week period and in four one-to-one sessions.

## Training of the intervention providers

The research nurse delivering the intervention did not have prior practice experience in rheumatology or allied specialties such as orthopaedics, rehabilitation or sports medicine. This further necessitated training in delivering treatments for arthritis and musculoskeletal diseases. Training was delivered by an academic physiotherapist (MH) and rheumatologist (AA) and focused on the assessment and management of OA in accordance with NICE guidelines, exercise prescription (aerobic and strengthening), information and advice to support weight loss and use of behavioural strategies to motivate patients and enhance adherence. Training

sessions were delivered in face-to-face sessions and supported by a training manual, case-studies and patient simulation sessions.

## Sample size

We aimed to recruit between 15 and 20 participants in the package development phase as this was expected to be sufficient to achieve data saturation. Saturation takes place when no new information is observed in the data and is estimated to occur after the conduction of twelve interviews [25]. Given the risk of drop-outs and participants not attending for the interview visit we aimed to recruit 15 to 20 participants in the study.

## Data collection

Participants in the study and the research nurse who delivered the intervention were invited to participate in a semi-structured interview. Participants were interviewed immediately after the end of their final treatment session. The nurse was interviewed at the end of the study and an additional interview was conducted via video call with the nurse after initial data analysis 45 weeks later, to explore any gaps or areas of uncertainty. Interviews were conducted using a narrative approach [26].

Two semi-structured interview guides, one each for the research nurse [21] and for people with knee pain (S1 File) were developed. The interview guide for people with knee pain covered: perceptions of disease management before, during, and after the nurse-led intervention; changes in perceptions of knee pain after the study; their experience of the intervention, the provider and delivery of the intervention; lifestyle changes; and, overall satisfaction with the treatment. The research nurse interview guide covered topics on their views and experiences of delivering the non-pharmacological intervention; confidence in delivering the package; and, possible barriers to implementing the package of care in clinical practice.

## Data analysis

The theoretical framework for assessing treatment acceptability was adapted from those published in previous literature [27] and guided the evaluation of acceptability in this study, with burden, ethicality, and intervention coherence being the key areas that represent acceptability of an intervention. Data were analysed using the framework approach [28]. All interviews were audio-recorded and transcribed verbatim by an external transcription company. Following this, transcripts were imported to NVivo 12 for analysis. Transcripts were reviewed and checked for accuracy, and any personal identification removed. Transcripts were read several times by PAN and segments of text coded. AF read a sample of three patient interviews and independently coded the transcripts. PAN and AF discussed initial codes, themes and sub-themes, which resulted in a working analytical framework. Codes identified in the nurse interview and patient interviews were similar and thus these were analysed together. The framework was then applied and refined following analysis of the remaining transcripts by PAN, and through discussion with the wider research team. Data were then indexed according to the final analytical framework and charted according to each theme, which facilitated data synthesis and interpretation.

## Results

Eighteen people with knee pain were recruited into the package development phase. Their mean (SD) age was 68.7 (9.0) years, 34% were women and the mean body mass index (BMI) was 31.2 (8.4) kg/m$^2$. Based on their BMI, nine were categorised as overweight and seven

**Table 1. Demographic details of participants according to whether they attended or did not attend all treatment visits.**

| Participant demographics | Attenders (N = 14) | Non-attenders (N = 4) |
|---|---|---|
| ✝ Age (years) | 69.8 (9) | 64.8 (6) |
| ✝ BMI (kg/m2) | 32.3 (9) | 27.5 (1) |
| Females (%) | 21 | 75 |
| Retired (%) | 86 | 100 |
| Knee pain severity (%) | | |
| Mild | 36 | - |
| Moderate | 29 | 100 |
| Severe | 35 | - |

✝ Values shown are mean (SD

obese. Seventeen out of eighteen were interviewed. including three of the four who did not attend all treatment visits. One participant was not contactable after the baseline visit and dropped out from the study. This participant did not participate in the interview. Participants, who did not attend all treatment visits were mostly females (75%) with mean (SD) age 64.8 (6) years and a lower BMI 27.5 (1) kg/m$^2$ compared with those who completed all visits. All non-attenders had moderate knee pain severity (see Table 1).

The average length of the interviews was 54.8 minutes (range 25–84 minutes). Analysis of the participants' accounts of their experiences led to the identification of three main themes (see Table 2).

## Theme 1: Participants' perception of the package of care

**Overall perception of the treatment package.** Most people with knee pain were satisfied with the package.

*"It has been very beneficial, because it made me think about doing different exercises and then when I did them, I can feel those working different muscles around the knee." (Patient 01)*

**Table 2. Themes and subthemes emerging from the interviews.**

| Themes | Subthemes |
|---|---|
| Participants' perception of the package of care | Overall perception of the package |
| | Perception of nurse-led care |
| | Perception of weight loss strategies |
| | Perception of the exercises |
| Raised awareness of the self-management of knee pain | Knowledge before the intervention |
| | Knowledge generation |
| | Knowledge reinforcement |
| | Perception around the Versus Arthritis UK booklet |
| | Impact of new knowledge |
| Engagement of patients to the intervention | Engagement to the advice in the development phase |
| | Psychological factors |
| | Trial participation |
| | Involvement of significant others |
| | Accountability to the nurse |
| | Perceived ability to continue with the advice |

They felt the intervention was informative overall, however, they thought that information given in the first session could be better spread out across two or three sessions.

"*I would say if it had been eased in a bit more, two or three sessions as opposed to the one that first week.*"*(Patient 02)*

**Perception of nurse-led care.**   People with knee pain appreciated that the nurse provided the opportunity and time to talk things through, which they did not always get with their GP or physiotherapist.

"*It is the first time that I have actually had the time to talk about them without feeling that I have to get in and out of a room.*" *(Patient 03)*

Most felt that the nurse used lay language and explained, demonstrated and corrected their exercises to good effect. They stated that the nurse was a good listener and communicator, attentive and knowledgeable, and felt that she was well suited to deliver this type of treatment. The nurse, built rapport, which created a trustworthy therapeutic relationship.

One participant discontinued the study due to dissatisfaction with the treatment package. The participant felt that the nurse should have focused more on exercises as other components of the package were "*common knowledge*". *(Patient 04)*

At the start of the study, the nurse reported low confidence in the ability to deliver the treatment, particularly with respect to selecting and prescribing exercises. This improved as the study progressed.

The nurse did not feel comfortable with goal setting. During goal setting, people with knee pain set difficult goals for themselves and rated their ability to achieve these goals as low. In this background, the nurse found it challenging to negotiate realistic goals and in motivating them. However, the nurse was able to do so in most instances.

**Perception of weight loss strategies.**   Most people with knee pain were satisfied with the weight loss advice. Measuring calorie content on foods made them realise how much they were eating and cut down on high calorie foods. Many acknowledged that reading food labels was challenging but very important to realise what food ingredients were consumed. Many were able to bring positive changes to their diet. This included switching from alcoholic drinks to non-alcoholic beverages and, from sweets to fruits. Others stopped buying ready meals from the supermarkets. For a few, the intervention required just small adjustments to their existing routines.

However, not all accepted weight loss advice. One felt there was no need to cut down on food at "*their age*" *(Patient 05)* while another who engaged with weight loss advice was confused with conflicting messages on calorie content from different sources. Another person who did not attend all treatment visits was reluctant to follow a low-calorie diet and to have their weight checked.

The nurse was aware discussing weight could be sensitive but felt comfortable providing weight loss advice.

"*I didn't feel uncomfortable but you need to kind of have a very fine line that someone doesn't get offended by your comment.*" *(Nurse)*

Despite this, a couple of participants felt aggrieved when advised to lose weight.

"*There was no kind of need to explain to me that I need to lose weight, well I was a bit annoyed because I have lost half a stone.*" *(Patient 04)*

**Perception of exercises.**   Many felt the nurse prescribed too many exercises on their first session, which caused them some physical discomfort. Although most found the exercises painful and tiring in the beginning, as they continued to do them daily, their pain became less frequent and less intense.

"*A couple of the exercises that I had to begin with . . .were quite painful to be honest. . .the pain is getting easier, less frequent, less intense.*" *(Patient 06)*

A few people with knee pain reported that following the nurse's advice positively influenced their mood and made them feel a lot happier because they were not getting out of breath as they used to.

Some found the exercises time consuming and felt that the nurse was too prescriptive in setting the required time commitment for them to implement the treatment regime, which they found difficult to fit into their day to day life.

"*She is very much keen on the idea that you should set aside, you should say, "This is your exercise time" and you go all the way through everything.*" *(Patient 07)*

Retired participants raised concerns that people in employment might find it difficult to fit the recommended exercise regime in their daily life due to other time commitments.

Most acknowledged that exercise sheets aided performing the exercises at home, as they served as a reminder of what the nurse had demonstrated to them during the research sessions. However, the lack of detailed verbal description within the exercise sheets made some feel uncertain as to how to perform them at home. They felt that exercise sheets could have a more in-depth explanation of the body position and how to perform the exercises. The nurse also felt that the exercise sheets could have been more descriptive and included video.

"*Video would have been helpful for them*". *(Nurse)*

## Theme 2: Raised awareness on the self-management of knee pain

**Knowledge before the intervention.**   People perceived knee pain to result from wear and tear that would get worse with ageing with no treatment available and the only option for improvement would be joint replacement surgery. Most were resigned to the idea that their condition would continue to get worse.

"*I did not know, that doing the exercises would improve, I thought the only solution would be an operation so that is what I have learned, or one of the things I have learned. . .I was kind of resigned to it getting worse.*" *(Patient 08)*

**Knowledge generation.**   Individualised education about OA changed patients' perception of the condition and they realised that worsening of symptoms is not inevitable (see later). Many reported gaining new knowledge about the calorie content of foods, which made them realise whether they were eating healthily or not. Patients also learnt about the importance of getting out of breath during their activities of daily living (e.g., walking) or during any exercise, and increasing their heart rate to achieve some aerobic activity. Some became aware of the importance of building muscle around the knee joint and losing weight, which would help them alleviate their pain and improve their condition.

"*I did not know exercises would help before, no, I thought it was worn out! The muscles they have gone and that was it but. . . then I was told that they could, you know, by exercises, be strengthened and improved.*" *(Patient 05)*

**Knowledge reinforcement.** Others were already aware of the benefits of exercise and weight loss in helping them to manage their knee pain. For those, the package worked as a good reminder and reinforced the knowledge of diet and exercise for OA rather than providing new knowledge.

"*This has served as a reminder as opposed to teaching me anything if you like. She's reinforced what I already thought about exercise and diet.*" *(Patient 06)*

**Perception around the Versus Arthritis UK booklet.** Many participants felt the booklet worked as a complementary element for the intervention, reinforcing information that they already knew about OA, and acting as a reminder for the exercises they needed to do.

"*It reinforced things I already knew. . . The book was essential, as far as me remembering the exercises and what she said about the exercises and that the basic ex- exercises in the book.*" *(Patient 08)*

**Impact of new knowledge.** Most participants reported that the package changed their perception of management of knee pain, understanding that it can be improved through self-management.

"*This package it's kind of changed my attitude really and if I can improve things muscle wise and mobility wise, then things could get better*". *(Patient 09)*

For those participants, the package increased motivation to perform exercises and achieve weight loss. Others started reading food labels and measuring their portion sizes because of the information learnt. Participants were hopeful their knee pain would improve if they implemented the advice given.

"*I'm thinking I may be able to keep it as it is not worse, I may be able to improve it slightly. . .which again would be a bonus*". *(Patient 10)*

## Theme 3: Engagement with the intervention

**Engagement to the advice in the development phase.** Those who were able to set time aside and establish a routine to perform their exercises were able to adopt to the exercise advice given by the nurse. Participants who were already exercising incorporated the exercise regime into their gym routine and performed the knee-targeted exercises at the gym.

"*I've sort of more or less developed a schedule to do these things. . . it has become part of my routine*". *(Patient 05)*

However, some found it challenging to find time to fit in the exercise regime, as they did not have a specific timeframe/routine to do the exercises. Others stated that adding more and more exercises to their regime can be time-consuming and will make them feel bored.

"*To add on, it's time consuming and, I think that can make you get bored, so I'm quite happy where it is really*". *(Patient 11)*

One participant who only attended two treatment sessions found it difficult to keep up with the exercise and diet regime. As a result, the participant opted instead to do his own exercise regime.

*"I am doing it in my own way, but not as intense as what the nurse wanted me to do. I've got a thing about physiotherapy, I don't really know, whether it works or not". (Patient 12)*

**Involvement of significant others.** Another factor that aided engagement to the exercise and/or weight loss for some was the involvement of partners or friends. During their home exercise performance, family members helped them in setting up exercise equipment (Thera-bands) or assisted with counting exercise sets. Family members were also involved in the weight loss regime and encouraged them to follow the weight loss programme. Others involved neighbours with knee pain and compared notes with them.

*"I was just walking by the garden and he [Neighbour] says I have been away on a course it's to do with arthritis of the knee. I said, well I'm on this so . . . I'll copy you my exercises and you have a go and see how you get on and every now and again we'll compare notes and, and see how we are and support each other". (Patient 07)*

**Psychological factors.** Commitment and self-discipline to perform the exercises and follow the weight loss advice in the belief that these would improve their knee pain also drove several patients to follow the advice. Filling the food diaries was a good discipline and encouraged the patients to follow the advice.

*"It's no use just ticking the boxes and pretending that doesn't help anybody, if you want to make an improvement then, you've got to do it, you've got to be committed to it." (Patient 13)*

Motivation and willingness to change also influenced their engagement with the intervention. Participants engaged more with the intervention after the nurse combined exercise and weight loss and realised their benefits.

*"This was a big change, and joining everything up for me, that is what has made it positive and succeed for me. . . being shown everything joined up and realising there might be a bit of light at the end of the tunnel" (Patient 14)*

One participant reported he was not motivated to follow all of the advice because his knee pain was only mild, and had his pain been greater he would have been more motivated to do so. The same person perceived the intervention as a complete lifestyle change and stated that this was not worth doing for what he would get back.

*"I mean that would have been a good, better motivation to carry on, had it been excruciating. . .You have to be motivated and it was a payoff between my level of pain and the amount that I had to do to off balance it."(Patient 12)*

**Trial participation.** Other participants who followed the advice of the nurse, reported doing so because they wanted to help with the research programme.

*"It was the participation, because if I'm doing it for myself, before I came here, I was quite happy with myself. I was managing, and it wasn't a major problem in my life, because I accepted the pain." (Patient 15)*

**Accountability to the nurse.** Having someone to be accountable to also drove several participants to adhere to the advice provided. As the nurse weighed them and checked on their exercise progress during follow-up treatment sessions, many followed the programme as they felt accountable to her.

"*Somebody was going to be measuring it and somebody was going to be saying, you've lost this much or you've stayed the same or you've put some on". (Patient 06)*

However, one participants that dropped out was worried that his compliance in the study would not be satisfactory, as he did not want to disappoint the nurse.

"*I didn't really want to let her down, you know, I thought well you can get somebody else who's going to be more compliant, you're going to get a better result, you know." (Patient 12)*

**Perceived ability to continue with the advice.** Even though participants were quite confident and motivated to continue and follow the advice in the programme, they felt accountable to the nurse to follow the exercise and weight loss advice and would have liked another session, at a later date, for the nurse to follow their progress.

"*A follow up in say three months, it's helpful, you know. I am okay to be motivated by myself, but to know that you would know that, that it continued, I just think that would be quite nice."(Patient 15)*

Having built a routine to do the exercises over the five-week programme, some participants felt they could continue to follow the advice in the long-term. Those who involved family members in the programme also felt accountable to them. Others reported that they would continue to use the diaries to help them carry on following the advice, and gradually increase the repetitions of the exercises.

"*I've got some diaries to carry on filling in and I will tweak the exercises slightly, increase the reps or whatever else, or the loading" (Patient 01)*

Having noticed the benefits of the programme on mobility and weight loss, many felt quite confident that they would continue with it.

"*I am going to follow on with both; I am going to carry on doing them while am getting the benefit. . . I don't see why I wouldn't ever get any benefit". (Patient 10)*

## Discussion

This study assessed the acceptability of a non-pharmacological package of care for knee pain by exploring nurse and participants' views and experiences of delivering and receiving the treatment. In this study, a trained research nurse was the sole contact with participants and managed the core non-pharmacological treatments independently. We assessed both post-intervention (retrospective) and anticipated (prospective) acceptability based on their views and experiences of what it would be like to continue with the intervention as part of an on-going routine care. The study followed guidelines for acceptability assessment [27] and assessed acceptability across intervention coherence and burden.

Intervention coherence refers to the extent to which participants understand the intervention and how it works. In the present study, most participants understood the commitments

they would be required to make and acknowledged how the intervention might work: via exercise, weight loss and better understanding of the disease process. Not unexpectedly, most people with knee pain stated that the most challenging part of the intervention was to fit the exercise regime into their daily routine.

Intervention burden reflects the reasons for discontinuing the study and the perceived amount of effort that is required to participate. Although the overall trial recruitment (from which this study sample was drawn) was small (n = 18), only four participants did not attend for all treatment visits. Two people who left the study stated that they were not able to make the required lifestyle changes or find time for doing the exercises at home.

For such complex interventions to be effective, the use of strategies to motivate patients and support adherence to healthy behaviours are also required [29, 30]. The nurse was able to implement them for the majority of participants. The treatment package changed patients' perception of managing knee pain by improving their understanding of nature of OA and rationale behind the use of non-pharmacological interventions and its management. This was done by individualised discussion on the nature of OA, risk factors for its onset and progression, mechanism of action of non-pharmacologic interventions, and the fact that OA symptoms improve with exercise and/or weight-loss in the short-term and potentially also in the long-term.

Overall, the vast majority of participants attended all treatment visits. However, the participants that did not attend treatment visits were younger, mostly female, and had lower BMI. Given the small sample size firm inferences cannot be drawn from this observation and this finding should be explored further in the pilot RCT. Key action points on how to modify and optimise the package of care in preparation for the feasibility RCT were identified. These include building in greater flexibility in the treatment package according to the individual needs of the patients, i.e., greater emphasis on exercises for some while greater emphasis on dietary and lifestyle changes for others, and further nurse training on goal setting and linking the goal to the exercises prescribed. Some participants found it difficult to practice exercise with the exercise sheets. We will therefore offer ad-hoc participant-initiated phone consultation to discuss the exercises with the nurse in the pilot-RCT. The findings reported in this manuscript were from the package development phase of research completed prior to the Covid-19 pandemic. It is likely that a future-RCT will utilise a hybrid model with a mix of face-to-face and online delivery of the complex package of care for knee pain.

Strengths of this study include community-based recruitment, achievement of data saturation, and cross-validation of data analysis by a trained qualitative researcher (AF). Medical Research Council (MRC) guidelines for developing and evaluating complex interventions stress the importance of collecting rich and targeted qualitative data to identify potential barriers and facilitators of intervention delivery [31], which we achieved through semi-structured interviews with inclusion of patients who did and did not complete all treatment sessions.

However, there are some caveats to the study. Firstly, the intervention was delivered by a single trained research nurse, and the study was carried out in a research setting. The results therefore may not be generalisable across real-world clinical settings and with intervention providers from different clinical backgrounds e.g. physiotherapists, dieticians. Evaluating only one nurse makes the findings of this study conditional on the ability of the nurse to deliver a complex package of care, implement behaviour change, use motivational techniques and negotiate action planning. It is possible that this may vary from one nurse to another. However, to minimise this effect we trained the research nurse in a structured manner, and chose a nurse to deliver the intervention with no prior experience of treating musculoskeletal disorders or utilising behaviour change techniques. The nurse was able to deliver the intervention with reasonably good acceptability and fidelity [21] suggesting that this complex package of care may

be deliverable by the vast majority of nursing healthcare professionals. This increases the potential for transferability of the findings of this research study to clinical settings. Secondly, the fidelity of intervention delivery during the study was assessed using video-recordings of individual treatment sessions [21], awareness of which may have affected the interviewer's line of questioning with participants. However, any risk of bias or judgement in questioning was minimised by conducting interviews at least one week after fidelity assessments and with the use of a pre-specified semi-structured interview.

In conclusion, nurse-led delivery of a non-pharmacological package of care for knee pain is acceptable in a research setting from both nurse and participant perspectives. However, there was just one research nurse delivering the intervention, and acceptability may be different for healthcare professionals in other settings. Most people with knee pain were satisfied with the package and found the advice supplied straightforward. The package changed their perception of managing knee pain, understanding that it can be improved though self-management. The results of the study are promising and support incorporation of the non-pharmacological package of care with minor modification into a feasibility RCT.

## Supporting information

**S1 File. Interview guide for participants.**
(PDF)

## Author Contributions

**Conceptualization:** Polykarpos Angelos Nomikos, Michelle C. Hall, Ana M. Valdes, Michael Doherty, David Andrew Walsh, Roshan das Nair, Abhishek Abhishek.

**Data curation:** Polykarpos Angelos Nomikos.

**Formal analysis:** Polykarpos Angelos Nomikos.

**Funding acquisition:** Abhishek Abhishek.

**Investigation:** Polykarpos Angelos Nomikos, Michael Doherty, Abhishek Abhishek.

**Methodology:** Polykarpos Angelos Nomikos, Michael Doherty.

**Software:** Polykarpos Angelos Nomikos.

**Supervision:** Michelle C. Hall, Amy Fuller, Ana M. Valdes, Roshan das Nair, Abhishek Abhishek.

**Validation:** Amy Fuller, Roshan das Nair.

**Writing – original draft:** Polykarpos Angelos Nomikos.

**Writing – review & editing:** Polykarpos Angelos Nomikos, Michelle C. Hall, Reuben Ogollah, Ana M. Valdes, Michael Doherty, David Andrew Walsh, Roshan das Nair, Abhishek Abhishek.

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
