## [Decision Letter · Decision Letter 0]

2 Aug 2021

PONE-D-21-07533

Acceptability of a nurse-led non-pharmacological complex intervention for knee pain: Nurse and patient views and experiences

PLOS ONE

Dear Dr. Polykarpos,

Thank you for submitting your manuscript to PLOS ONE. After careful consideration, we feel that it has merit but does not fully meet PLOS ONE’s publication criteria as it currently stands. Therefore, we invite you to submit a revised version of the manuscript that addresses the points raised during the review process.

As an editor, I feel this topic is interesting, releavnt and timely to osteoarthritis care.  Reviwers 1 and 2 raised important similar points which I agreed should all be responded to and where appropriate make chamnges to the manuscript.

We look forward to receiving your revised manuscript.

Kind regards,

Ukachukwu Okoroafor Abaraogu

Academic Editor

PLOS ONE

Journal Requirements:

Reviewers' comments:

Reviewer's Responses to Questions

**Comments to the Author**

1. Is the manuscript technically sound, and do the data support the conclusions?

Reviewer #1: Yes

Reviewer #2: Yes

2. Has the statistical analysis been performed appropriately and rigorously? 

Reviewer #1: N/A

Reviewer #2: I Don't Know

3. Have the authors made all data underlying the findings in their manuscript fully available?

Reviewer #1: Yes

Reviewer #2: Yes

4. Is the manuscript presented in an intelligible fashion and written in standard English?

Reviewer #1: No

Reviewer #2: Yes

5. Review Comments to the Author

Reviewer #1: Thanks for the opportunity to review the manuscript. This is a feasibility study for a follow-up randomised-controlled trial. The aim was to evaluate the acceptability of a nurse-led non-pharmacological package of care for knee osteoarthritis. The topic is interesting with a good introduction. However, certain aspects of the manuscript need to be clarified. As an improvement, I have made suggestions bellow.

The Abstract and Introduction is satisfactory.

METHODS - The author is likely to benefit from writing in shorter sentences for clarity. The readers may lose the message in the paragraph as it sometimes loses its meaning.

#1. Consider using this format for the method section. First and foremost, describe the method used. Study design... Was ethical approval sort? Consent from participants? Patient and Public Involvement? Please, describe the expertise, background and training of the intervention providers (e.g. number of therapists, professions, expertise). Then use the standard recruitment/participants, settings, intervention, and data collection and analysis.

#2. Lines 92-97. Please consider rewriting.

“Respondents to this survey were individuals aged 40 and over with knee pain for more than 3 months and pain in or around the knee on most days of the previous month. Participants with at least moderate pain in two of the five domains of the Western Ontario and McMaster Universities Osteoarthritis Index (WOMAC) pain scale were invited to participate in the study [22]. Study participants were also limited to individuals who can read and write in English.”

#3. Lines 103-106. Please consider rewriting.

“The research nurse had no prior practice experience in rheumatology or allied specialties such as orthopaedics, rehabilitation, or sports medicine. This further necessitated training in delivering treatments for arthritis and musculoskeletal diseases.”

#4. Lines 106-111. Consider rewriting.

The nurse delivered the non-pharmacological package of care over a five-week period and in four one-to-one sessions which comprises (i) holistic assessment; (ii) patient education and advice; (iii) aerobic and strengthening exercise and (iv) weight-loss advice where required. Advice on the use of adjunctive treatments such as the application of heat or cold foot-wear modification and the use of walking aids were discussed in the... session (please name the session).

#5. Lines 116-120 should be discussed as described in #1

#6. Line 116. Sample size. Although it was clear that this is a pilot study, how did the authors come about the sample size used for the study.

There are several literature on suitability of sample size for various research. Please see Charan and Biswas (2013) article on “How to calculate sample size for different study designs in medical research?”

#7. Line 121. Please rename section “Interview study” to “Data collection”

#8. Line 126. “(PhD student and physiotherapist)” is not necessary. Please delete. This should have been discussed as described in #1

#9. Line 126. “The research nurse was approached by PAN to participate in an interview”. This is not clear. Please consider removing.

#10. Line 127. “A theoretical framework for assessing treatment acceptability has been published [24]”. Consider rewriting as “The theoretical framework for assessing treatment acceptability was adapted from those published in previous literature [24].”

Lines 133-135 is not required. See #1

Lines 144-145. “The research nurse interview was conducted by PAN and AF, one week after the final treatment session for the last patient”. Again, see #1.

Lines 148-151. This should be discussed as described in #1. I believe that informed consent has been discussed already in line 101. The authors need to be concise.

Lines 151-153. Again this already appeared in line 98 “setting”. This should have been discussed entirely elucidated in settings. Please see #1.

Lines 153-158. See #1 (Please, describe the expertise, background and training of the intervention providers (e.g. number of therapists, professions, expertise).

RESULTS

Line 304: “Perception of the Arthritis UK booklet”. Please make corrections to table 1 “Perception around the Versus Arthritis UK booklet”.

DISCUSSION

Line 456: “Firstly, the intervention was delivered by a single trained research nurse, and the study was carried out in a research setting”. So what impact does it have on the study?

Line 458: “the participants were recruited from a primary-care based cohort. These factors increase the transferability of the findings to clinical settings” The author already discussed the strength of the study. Is this meant to be a strength or limitation?

Line 460: Please remove “the main interviewer (PAN)”. You can start the sentence with “The fidelity of intervention delivery during the study was assessed using video-recordings of individual treatment sessions.”

ACKNOWLEDGMENTS

This section presently does not really clearly describe any funding or input from any expert that was not included as an author. Suggest you replace with “The authors declare no source of funding” or use text recommended by the journal.

Reviewer #2: Congratulations to all co-authors on this really interesting feasibility study, it was a pleasure to read. This piece of work greatly contributes to an area of research which is poorly understood – particularly as so much prominence is put on pharmacological based interventions.

The framework analysis is not my area of expertise, but I have identified some minor points for clarification:

1. Pg 7, L111 “Advice on the use of adjunctive treatments such as the application of heat or cold, foot-wear modification and the use of walking aids were discussed.”

Could more detail be added to describe the types of footwear modification/ type of walking aids?

2. In the results/discussion, more detail is required on how the intervention changed patients’ perception of managing knee pain – it is unclear how this is nurse-led package of care is acceptable in the clinical settling.

3. The feasibility study could have been enhanced by delivery of more than one-nurse led non-pharmacological intervention, mentioned in limitations but also important that this is taken into consideration when interpreting results. For example, the nurse found discussion of goal setting to be challenging – another practitioner may have found this easier. Please expand on the limitation of evaluating only one nurse-practitioner, as level of experience of delivering behaviour change techniques and familiarity with knee OA and exercise provision will ultimately affect intervention delivery.

4. Pg4, L64 First use of ‘GP’ abbreviation in paper

5. Pg6, L103 Request more details on how the nurse was trained to deliver the intervention? Did they have previous experience of using behaviour change techniques?

6. Pg 23, L445 Have they considered other means of delivering the exercises i.e. using web-based/mobile health which have been used in other areas e.g. inflammatory arthritis?

7. Please clarify: 18 participants were recruited for study, only 17 participated in the interview? This sentence in results section may need restructured to reflect this, and if possible an explanation of exclusion of one participant?

8. There was no detail on the comparison of participants who attended all the sessions Vs those who did not. Could any difference be discussed?

6. PLOS authors have the option to publish the peer review history of their article (what does this mean?). If published, this will include your full peer review and any attached files.

Reviewer #1: No

Reviewer #2: No

---

## [Author Response · Author response to Decision Letter 0]

24 Sep 2021

RESPONSE TO EDITOR AND REVIEWERS

Response to the editor:

Dear Dr Ukachukwu Okoroafor Abaraogu, 

We thank you and the reviewers for their insightful and helpful comments. We would like to inform you that the edits (including page and line numbers) correspond to the original version of the manuscript and not the revised version with the track changes.

EDITOR COMMENT: As an editor, I feel this topic is interesting, relevant and timely to osteoarthritis care. Reviewers 1 and 2 raised important similar points which I agreed should all be responded to and where appropriate make changes to the manuscript.

REVIEWER COMMENT: Reviewer #1: Thanks for the opportunity to review the manuscript. This is a feasibility study for a follow-up randomised-controlled trial. The aim was to evaluate the acceptability of a nurse-led non-pharmacological package of care for knee osteoarthritis. The topic is interesting with a good introduction. However, certain aspects of the manuscript need to be clarified. As an improvement, I have made suggestions bellow.

The Abstract and Introduction is satisfactory.

METHODS - The author is likely to benefit from writing in shorter sentences for clarity. The readers may lose the message in the paragraph as it sometimes loses its meaning.

Comment #1. Consider using this format for the method section. First and foremost, describe the method used. Study design... Was ethical approval sort? Consent from participants? Patient and Public Involvement? Please, describe the expertise, background and training of the intervention providers (e.g. number of therapists, professions, expertise). Then use the standard recruitment/participants, settings, intervention, and data collection and analysis.

Response: We thank the reviewer for their insightful comments. We have now altered the sequence of methods and presented as suggested. Please see pages 7-8, lines 127-137 for the description of the expertise, background and training of the intervention providers.

Comment #2. Lines 92-97. Please consider rewriting. “Respondents to this survey were individuals aged 40 and over with knee pain for more than 3 months and pain in or around the knee on most days of the previous month. Participants with at least moderate pain in two of the five domains of the Western Ontario and McMaster Universities Osteoarthritis Index (WOMAC) pain scale were invited to participate in the study [22]. Study participants were also limited to individuals who can read and write in English.”

Response: We thank the reviewer for their suggestion. We have now re-written the section. Please see page 6, lines 102-107. 

 

Comment #3. Lines 103-106. Please consider rewriting. “The research nurse had no prior practice experience in rheumatology or allied specialties such as orthopaedics, rehabilitation, or sports medicine. This further necessitated training in delivering treatments for arthritis and musculoskeletal diseases.”

Response: We thank the reviewer for their suggestion. We have now re-written the section. Please see pages 7-8, lines 127-131.

Comment #4. Lines 106-111. Consider rewriting. The nurse delivered the non-pharmacological package of care over a five-week period and in four one-to-one sessions which comprises (i) holistic assessment; (ii) patient education and advice; (iii) aerobic and strengthening exercise and (iv) weight-loss advice where required. Advice on the use of adjunctive treatments such as the application of heat or cold foot-wear modification and the use of walking aids were discussed in the... session (please name the session).

Response: We thank the reviewer for their suggestion. We have now re-written the section. “The intervention consisted of a holistic assessment; education and advice; prescription of aerobic and strengthening exercise and weight-loss advice where required as recommended by NICE guidelines and is described in more detail in the protocol [20]. Specific advice around the use of adjunctive treatments such as the application of heat or cold, foot-wear modification and the use of walking aids were included although not allocated to individual sessions. Please see page 7, lines 112-117.

Comment #5. Lines 116-120 should be discussed as described in #1. 

Response: We thank the reviewer for their suggestion. We have now re-written the section. Please see page 8, lines 131-137 and page 6, lines 95-97.

Comment #6. Line 116. Sample size. Although it was clear that this is a pilot study, how did the authors come about the sample size used for the study.

Response: We thank the reviewer for their suggestion. We have now provided justification for the sample size. Please see page 8, lines 138-143.

Comment #7. Line 121. Please rename section “Interview study” to “Data collection”

Response: We thank the reviewer for their suggestion. We have now renamed the section of “Interview study” as “Data collection”. Please see page 8, line 144

Comment #8. Line 126. “(PhD student and physiotherapist)” is not necessary. Please delete. This should have been discussed as described in #1

Response: We thank the reviewer for their suggestion. We have now deleted “(PhD student and physiotherapist)”. 

Comment #9. Line 126. “The research nurse was approached by PAN to participate in an interview”. This is not clear. Please consider removing.

Response: We thank the reviewer for their suggestion. We have now removed “The research nurse was approached by PAN to participate in an interview”.

Comment #10. Line 127. “A theoretical framework for assessing treatment acceptability has been published [24]”. Consider rewriting as “The theoretical framework for assessing treatment acceptability was adapted from those published in previous literature [24].”

Response: We thank the reviewer for their suggestion. We have now revised the sentence as requested and included the theoretical framework in the data analysis section. Please see page 9, lines 161-164. 

Comment #11. Lines 133-135 is not required. See #1

Response: We thank the reviewer for their suggestion. We have now deleted the section as requested. 

Comment #12. Lines 144-145. “The research nurse interview was conducted by PAN and AF, one week after the final treatment session for the last patient”. Again, see #1.

Response: We thank the reviewer for their suggestion. We have now deleted the section as requested. 

Comment #13 Lines 148-151. This should be discussed as described in #1. I believe that informed consent has been discussed already in line 101. The authors need to be concise.

Response: We thank the reviewer for their suggestion. We agree that informed consent is described and have now deleted the section as suggested. 

Comment #14 Lines 151-153. Again this already appeared in line 98 “setting”. This should have been discussed entirely elucidated in settings. Please see #1.

Response: We thank the reviewer for their suggestion. We have now moved the section as suggested. 

Comment #15 Lines 153-158. See #1 (Please, describe the expertise, background and training of the intervention providers (e.g. number of therapists, professions, expertise).

Response: We thank the reviewer for their suggestion. We have now moved the section as requested. Please see pages 7-8, lines 127-137. 

RESULTS

Comment #16 Line 304: “Perception of the Arthritis UK booklet”. Please make corrections to table 1 “Perception around the Versus Arthritis UK booklet”.

Response: We thank the reviewer for noticing this detail. We have now corrected the relevant section of table 1. The corrections correspond to Table 2 please see page 11. 

DISCUSSION

Comment #17 Line 456: “Firstly, the intervention was delivered by a single trained research nurse, and the study was carried out in a research setting”. So what impact does it have on the study?

Response: We thank the reviewer for the suggestion. We have now added the impact of a single trained research nurse delivering the intervention in a research setting. Please see page 24, lines 473-476: “Evaluating only one nurse makes the findings of this study conditional on the ability of the nurse to deliver a complex package of care, implement behaviour change, use motivational techniques and negotiate action planning. It is possible that this may vary from one nurse to another”

Please see pages 24, lines 471-473: “The results therefore may not be generalisable across real-world clinical settings and with intervention providers from different clinical backgrounds e.g. physiotherapists, dieticians.” 

Comment #18 Line 458: “the participants were recruited from a primary-care based cohort. These factors increase the transferability of the findings to clinical settings” The author already discussed the strength of the study. Is this meant to be a strength or limitation?

Response: We thank the reviewer for the suggestion. We have now mentioned that this is a particular strength. Please see page 23, line 462: “Strengths of this study include community-based recruitment”

Comment #19 Line 460: Please remove “the main interviewer (PAN)”. You can start the sentence with “The fidelity of intervention delivery during the study was assessed using video-recordings of individual treatment sessions.”

Response: We have now removed that section and added, “The fidelity of intervention delivery during the study was assessed using video-recordings of individual treatment sessions” adding the relevant reference. Please see page 24, lines 483-486. 

ACKNOWLEDGMENTS

Comment #20 this section presently does not really clearly describe any funding or input from any expert that was not included as an author. Suggest you replace with “The authors declare no source of funding” or use text recommended by the journal.

Response: We thank the reviewer for the suggestion. The journal states: “Do not include funding or competing interests information in Acknowledgments.” However, we have added the relevant Acknowledgement section: 

“None declared”

REVIEWER COMMENT: Reviewer #2: Congratulations to all co-authors on this really interesting feasibility study, it was a pleasure to read. This piece of work greatly contributes to an area of research which is poorly understood – particularly as so much prominence is put on pharmacological based interventions.

The framework analysis is not my area of expertise, but I have identified some minor points for clarification:

Comment #1. Pg 7, L111 “Advice on the use of adjunctive treatments such as the application of heat or cold, foot-wear modification and the use of walking aids were discussed.”

Could more detail be added to describe the types of footwear modification/ type of walking aids?

Response: We thank the reviewer for the suggestion. We have now added: “Rather advice was tailored to the participants and the nurse advised on the wearing of footwear with a cushioned sole and the use of walking stick where participants walked with an antalgic gait. This is a more concise description of the intervention and further details are available in the study protocol.” Please see page 7, lines 117-121. 

Comment #2. In the results/discussion, more detail is required on how the intervention changed patients’ perception of managing knee pain – it is unclear how this is nurse-led package of care is acceptable in the clinical settling.

Response: We thank the reviewer for these comments. We have now added

“The treatment package changed patients’ perception of managing knee pain by improving their understanding of nature of OA and rationale behind the use of non-pharmacological interventions and its management. This was done by individualised discussion on the nature of OA, risk factors for its onset and progression, mechanism of action of non-pharmacologic interventions, and the fact that OA symptoms improve with exercise and/or weight-loss in the short-term and potentially also in the long-term.”

Please see page 23, lines 439-445. 

We have now acknowledged: “The intervention was delivered by a single trained research nurse, and the study was carried out in a research setting. The results therefore may not be generalisable across real-world clinical settings and with intervention providers from different clinical backgrounds e.g. physiotherapists, dieticians.”

Please see page 24, lines 469-473.

The nurse was able to deliver the intervention with reasonably good acceptability and fidelity suggesting that this complex package of care may be deliverable by the vast majority of nursing healthcare professionals. This increases the potential for transferability of the findings of this research study to clinical settings.

Please see page 24, lines 479-483.

Comment #3. The feasibility study could have been enhanced by delivery of more than one-nurse led non-pharmacological intervention, mentioned in limitations but also important that this is taken into consideration when interpreting results. For example, the nurse found discussion of goal setting to be challenging – another practitioner may have found this easier. Please expand on the limitation of evaluating only one nurse-practitioner, as level of experience of delivering behaviour change techniques and familiarity with knee OA and exercise provision will ultimately affect intervention delivery.

Response: We thank the reviewer for their helpful comment. We have now expanded on that particular limitation. Please see page 24, lines 473-476: “Evaluating only one nurse makes the findings of this study conditional on the ability of the nurse to deliver a complex package of care, implement behaviour change, use motivational techniques and negotiate action planning”.

Comment #4. Pg4, L64 First use of ‘GP’ abbreviation in paper

Response: We thank the reviewer for noticing this detail. We have now provided the full terminology of GP when firstly used. Please see page 4, line 65.

Comment #5. Pg6, L103 Request more details on how the nurse was trained to deliver the intervention? Did they have previous experience of using behaviour change techniques?

Response: We thank the reviewer for their helpful comment. We have now added: 

“Training was delivered by an academic physiotherapist (MH) and rheumatologist (AA) and focused on the assessment and management of OA in accordance with NICE guidelines, exercise prescription (aerobic and strengthening), information and advice to support weight loss and use of behavioural strategies to motivate patients and enhance adherence. Training sessions were delivered in face-to-face sessions and supported by a training manual, case-studies and patient simulation sessions.” Please see pages 7-8, lines 131-137.

Comment #6. Pg 23, L445 Have they considered other means of delivering the exercises i.e. using web-based/mobile health which have been used in other areas e.g. inflammatory arthritis?

Response: We thank the reviewer for their helpful comment. We have not considered training the nurse to deliver web-based exercises/mobile health and we have mentioned this for future research: 

“The findings reported in this manuscript were from the package development phase of research completed prior to the Covid-19 pandemic. It is likely that a future-RCT will utilise a hybrid model with a mix of face-to-face and online delivery of the complex package of care for knee pain.” Please see page 23, lines 457-461.

Comment #7. Please clarify: 18 participants were recruited for study, only 17 participated in the interview? This sentence in results section may need restructured to reflect this, and if possible an explanation of exclusion of one participant?

Response: We thank the reviewer for their helpful comment. We have now added:

Please see page 10, lines 182-185: “Seventeen out of eighteen were interviewed. including three of the four who did not attend all treatment visits. One participant was not contactable after the baseline visit and dropped out from the study. This participant did not participate in the interview. 

Comment #8. There was no detail on the comparison of participants who attended all the sessions Vs those who did not. Could any difference be discussed?

Response: We thank the reviewer for their helpful comment. We have now included Table 1 (page 10) and added: “Participants, who did not attend all treatment visits were mostly females (75%) with mean (SD) age 64.8 (6) years and a lower BMI 27.5 (1) kg/m2 compared with those who completed all visits. All non-attenders had moderate knee pain severity (see Table 1)”. Please see page 10, lines 185-188. 

Moreover, we identified potential barriers and facilitators of intervention delivery, which we achieved through semi-structured interviews with inclusion of patients who did and did not complete all treatment sessions. Please see page 22, lines 434-436: 

“Two people who left the study stated that they were not able to make the required lifestyle changes or find time for doing the exercises at home”

---

## [Decision Letter · Decision Letter 1]

26 Dec 2021

Acceptability of a nurse-led non-pharmacological complex intervention for knee pain: Nurse and patient views and experiences

PONE-D-21-07533R1

Dear Dr.  Nomikos,

We’re pleased to inform you that your manuscript has been judged scientifically suitable for publication and will be formally accepted for publication once it meets all outstanding technical requirements.

Kind regards,

Ukachukwu Okoroafor Abaraogu, BMR(PT), MSc, PhD

Academic Editor

PLOS ONE

Additional Editor Comments (optional):

Reviewers' comments:

Reviewer's Responses to Questions

**Comments to the Author**

1. If the authors have adequately addressed your comments raised in a previous round of review and you feel that this manuscript is now acceptable for publication, you may indicate that here to bypass the “Comments to the Author” section, enter your conflict of interest statement in the “Confidential to Editor” section, and submit your "Accept" recommendation.

Reviewer #1: All comments have been addressed

Reviewer #2: All comments have been addressed

2. Is the manuscript technically sound, and do the data support the conclusions?

Reviewer #1: Yes

Reviewer #2: Yes

3. Has the statistical analysis been performed appropriately and rigorously? 

Reviewer #1: N/A

Reviewer #2: Yes

4. Have the authors made all data underlying the findings in their manuscript fully available?

Reviewer #1: Yes

Reviewer #2: Yes

5. Is the manuscript presented in an intelligible fashion and written in standard English?

Reviewer #1: Yes

Reviewer #2: Yes

6. Review Comments to the Author

Reviewer #1: Thank you for the opportunity to review your manuscript. All peer-review suggestions have been addressed. The manuscript could still benefit from the journal's language editing prior to its publication.

Reviewer #2: (No Response)

7. PLOS authors have the option to publish the peer review history of their article (what does this mean?). If published, this will include your full peer review and any attached files.

Reviewer #1: No

Reviewer #2: **Yes: **Dr Joanne E Hurst

---

## [Editor Report · Acceptance letter]

5 Jan 2022

PONE-D-21-07533R1 

Acceptability of a nurse-led non-pharmacological complex intervention for knee pain: Nurse and patient views and experiences 

Dear Dr. Nomikos:

I'm pleased to inform you that your manuscript has been deemed suitable for publication in PLOS ONE. Congratulations! Your manuscript is now with our production department. 

Kind regards, 

on behalf of

Dr. Ukachukwu Okoroafor Abaraogu 

Academic Editor

PLOS ONE